# Effects of Physical Exercise on Symptoms and Quality of Life in Women in Climacteric: A Systematic Review and Meta-Analysis

**DOI:** 10.3390/healthcare13060644

**Published:** 2025-03-15

**Authors:** Providencia Juana Trujillo-Muñoz, María Angustias Sánchez-Ojeda, Eva Carolina Rodríguez-Huamán, Karima Mezyani-Haddu, Irene Hoyo-Guillot, Silvia Navarro-Prado

**Affiliations:** 1RN, Regional Hospital of Melilla, 52005 Melilla, Spain; ptm39@ugr.es (P.J.T.-M.); evrohu@correo.ugr.es (E.C.R.-H.); kmezyani@correo.ugr.es (K.M.-H.); 2Department of Nursing, Faculty of Health Sciences, University of Granada, 52005 Melilla, Spain; irenehoyog@ugr.es (I.H.-G.); silnado@ugr.es (S.N.-P.)

**Keywords:** menopause, postmenopause, climacteric, physical activity, aerobic exercise, strength training, quality of life, hot flashes, anxiety

## Abstract

**Background/Objectives:** Climacteric is a period of hormonal changes in women characterised by physical, emotional, and interpersonal alterations. This period is divided into two phases: perimenopause (the period from the appearance of the first symptoms to the arrival of menopause) and postmenopause (up to approximately 64 years of age). The progression of symptoms varies for each woman and can negatively affect self-esteem and quality of life. One of the most commonly used tools to reduce these negative effects is the performance of different types of physical exercise. The objective of this review was to summarise the research on the effects of both aerobic and strength exercises in women during the transition from perimenopause to postmenopause. **Methods:** This systematic review and meta-analysis was conducted according to the PRISMA 2020 guidelines. Initially, 1995 articles published in PubMed, Scopus and Web of Science between January 2014 and June 2024 were identified. From these articles, two researchers separately selected eight randomised controlled trials and compared the effects of aerobic and strength exercises with no activity. The risk of bias in the included articles was assessed using the Cochrane Risk of Bias tool for randomised clinical trials. **Results:** Most of the trials reported that both types of exercise, when performed in a controlled and regular way, have beneficial effects on the vitality and mental health of women in climacteric, increasing their general quality of life. A meta-analysis revealed that aerobic exercise improved the symptoms of menopause, although studies that support the effect of aerobic exercise on vasomotor symptoms are lacking. Resistance exercise was shown to strengthen muscles, increase bone density, and protect against osteoporosis. **Conclusions:** Physical exercise is a safe and nonpharmacological option that has positive effects on the health of women in climacteric.

## 1. Introduction

Climacteric is a period of hormonal change in women marked by physical, emotional and interpersonal alterations. This period is divided into two phases: perimenopause (the period from the onset of the first symptoms to the arrival of menopause), and postmenopause (up to approximately 64 years of age) [1]. The World Health Organization (WHO) defines menopause as the definitive cessation of ovarian activity without pathological or physiological cause and is diagnosed after 12 months of amenorrhea. Menopause usually begins around the age of 50, but some women may experience it early before the age of 40. On the other hand, menopause can occur later in life when it occurs after the age of 55 [2]. Its onset can be natural or caused by medical treatments [3]. Menopausal symptoms vary widely, and the most common symptoms are vasomotor symptoms (VMSs), such as hot flashes and night sweats; sleep difficulties and insomnia; decreased vaginal moisture; weight gain; sarcopenia; and altered mood [4,5,6]. These symptoms have a substantial impact on the daily lives of women and decrease their quality of life (QoL) [7,8]. In some cases, symptom progression can negatively affect a woman’s self-esteem, causing difficulties in both social and work relationships. Menopause is associated with increased suffering from certain chronic diseases, such as osteoporosis, cardiovascular diseases and metabolic conditions [9]. In addition, over time, certain types of pain, such as musculoskeletal pain or fibromyalgia, increase. To address menopausal symptoms, lifestyle changes, highlighting diet and exercise, are essential [10,11].

In addition to the physical and emotional changes associated with menopause, such as decreased bone density, redistribution of body fat, and its impact on cardiovascular health [12], many women experience mood disturbances, fatigue, and mild cognitive difficulties [2]. To mitigate these effects, various therapeutic strategies are available, including hormone replacement therapy and other pharmacological treatments, such as selective oestrogen receptor modulators and certain antidepressants. In addition, to increase the quality of life of women at this stage, physical exercise, a balanced diet, and rest are recommended [13,14]. However, despite the efficacy of these interventions, physical exercise has been demonstrated to be a nonpharmacological alternative with multiple benefits for the overall health and QoL of women during the climacteric period [7].

Because of the beneficial effects on the cardiovascular system, physical exercise is the first-line nonpharmacological treatment for the prevention of diseases in women in climacteric. Exercise increases physical fitness, which is strongly associated with better mental and metabolic health, and delays arterial ageing by maintaining adequate levels of blood pressure [15,16,17]. In this context, resistance training (RT) is widely recommended; however, participation in and adherence to these programs are usually low. Currently, elastic bands are among the most commonly used equipment in RT, because they are an economical option that is easy to use and adjustable to different intensities. The benefits of RT include increased muscle and bone mass, reduced sensitivity to pain, and improved QoL [18]. In addition, RT has a decisive effect on the frequency of VMSs, and its reduction in the impact of menopause continues to recommend its use [19]. Similarly, the Scientific Advisory Committee and Patient Information Committee of the Royal College of Obstetricians and Gynaecologists recommend regular and sustained aerobic exercise (AE), such as running, swimming, or hiking, as an effective intervention to maintain weight and control blood pressure, metabolism, and lipid levels, which also minimise symptoms in these women [20].

The tools used to assess health-related quality of life (HRQOL) can be classified into generic and specific instruments. Among the generic instruments, the Short Form-36 (SF-36) is one of the most frequently used instruments. The SF-36 consists of 36 items that assess HRQOL in eight areas, including functional status, satisfaction, and health analysis, and is used to measure the perception of health in the general population [21]. Some researchers have proposed the need for a questionnaire with fewer items to evaluate these dimensions; the SF-12 is the abbreviated version of the original scale with only 12 items [22]. The Women’s Health Questionnaire (WHQ) is used to measure HRQOL specifically during menopause. The WHQ consists of 36 items grouped into nine subscales that address somatic symptoms, depressive attitudes, memory and concentration limitations, anxieties and fears, sexual behaviour, VMSs, sleep impairment, menstrual disorders, and attraction [23]. Similarly, the Cervantes scale is used to evaluate menopause in the Spanish female population between 45 and 64 years of age. The Cervantes scale contains 31 items, with an abbreviated version with 15 items, classified into four dimensions: health in menopause, sexual life, emotional control, and relationships [24]. All of these instruments evaluate QoL by focusing mainly on the patient’s perception of their own health [23].

Although there are studies examining the impact of exercise on climacteric symptoms, the results may not always be conclusive. The need to determine the impact of different types of physical exercise, particularly aerobic and strength training, as well as optimal duration and intensity, may provide stronger evidence to guide clinical recommendations. The aim of this systematic review with meta-analysis of randomised clinical trials (RCTs) is to provide an update on the effects of AE and RT in women during the climacteric period to analyse the long-term effectiveness of these programmes.

## 2. Materials and Methods

### 2.1. Study Design and Selection Criteria

A systematic review was performed to synthesise the available evidence on the influence of physical exercise on the symptoms and QoL of climacteric women according to the PRISMA Declaration for Systematic Reviews and Meta-analysis [25]. The PICO framework was used to establish the eligibility criteria for the review and meta-analysis as follows:Patients (P): women in perimenopause, menopause, and postmenopause (stages included in climacteric), aged between 40 and 70 years. Studies of women with unnatural menopause or postmenopause and with medical conditions that could significantly influence physical activity or climacteric symptoms were excluded.Interventions (I): AE such as warm-up, walking, mobility, and stretching with relaxation and RT with machines and/or strengthening with elastic bands. The minimum required duration of the intervention was 12 weeks, with 1–3 sessions/week, to ensure sufficient time to observe significant changes in the health of the women. Studies focusing on physical activity combined with other interventions (diet and/or medication) where the effects of physical activity could not be isolated or mental activities such as yoga, tai chi, Pilates, etc., were excluded.Comparators (C): groups with no intervention, minimal intervention, or placebo (health education without physical activity) and groups with activity, compared according to QoL scores. Studies with groups with interventions that were not well defined were excluded.Outcomes (O): symptoms related to menopause, such as hot flashes, insomnia, and mood changes, among others, and HRQOL assessment using validated questionnaires (Menopause-specific Quality of Life [MENQOL], SF-36 and SF-12) and the Cervantes scale. Studies in which symptoms of menopause were not evaluated after physical activity were excluded.

The electronic databases PubMed, Scopus, and Web of Science were searched, along with reference lists of known relevant articles. RCTs published between January 2014 and June 2024 that were peer-reviewed and written in English or Spanish were evaluated. MeSH terms and a wide variety of related words were used in advanced searches. The MeSH terms included menopause, postmenopause, climacteric, physical activity, aerobic exercise, strength training, quality of life, hot flashes, and anxiety. The following search string was created using Boolean operators and proximity and truncation operators for effective recovery: ((“menopause” OR “post menopause” OR “climacteric”) AND (“physical activity” OR “aerobic exercise” OR “strength training”) AND (“quality of life” OR “hot flashes” OR “anxiety”)). All identified studies were stored in a bibliographic manager for later evaluation. Observational, cohort, and case–control studies; non-peer-reviewed studies; and studies without available full text were also excluded. After elimination of duplicates, two investigators (P.J.T.-M. and E.C.R.-H.) separately examined the titles, abstracts, and full texts of the studies to determine their eligibility. Any possible disagreements between the investigators’ findings were resolved by a third and fourth author (S.N.-P. and M.A.S.-O.) to reduce possible disagreements. The entire process was carried out independently according to the PICO criteria described above via reference management software (Zotero 5.0.) [26]. This review has not been registered and no prior protocol was developed, although it follows the PRISMA guidelines to ensure methodological rigour.

### 2.2. Data Collection and Publication Bias

The data extracted from the reviewed articles were recorded in an evidence table and grouped into 6 sections: section 1, author, year of publication, and region; section 2, randomisation of the control and intervention groups; section 3, number of participants and stage of menopause; section 4, number of intervention groups and control groups and frequency and total duration of the intervention; section 5, description of the intervention, RT (work with machines and elastic bands) and AE (walking, swimming, and Nordic walking); and section 6, methods used to measure QoL and climacteric symptoms. The table was created for the purpose of organising the information extracted from the articles reviewed. This information was then restructured into Table 1, to which the corresponding results were added for a clearer and more complete presentation. The risk of bias in the included RCTs was independently assessed by two researchers (P.J.T.-M. and E.C.R.-H.) using the second version of the Cochrane Risk of Bias tool (RoB 2) [27]. Any discrepancies in the risk of bias classification were resolved through consensus with a third author (K.M.-H.).

### 2.3. Statistical Analysis

Two meta-analyses were carried out using forest diagrams to calculate the effect size or relative risk of suffering climacteric symptoms according to whether physical exercise was practised, differentiating between AE and RT. The I^2^ value was used to assess heterogeneity; for an I^2^ value < 50%, a meta-analysis was performed using fixed effects. Egger’s test was used to assess publication bias, and a sensitivity analysis was performed. Data analyses were performed using Review Manager 5.4 software [33].

Study selection was carried out in four stages, as shown in the flow diagram in Figure 1. First, RCTs in English and/or Spanish that were published in the previous 10 years were identified, and duplicate articles were discarded, resulting in 183 articles. Second, the articles that met the eligibility criteria were selected according to the titles and abstracts, and 12 articles were identified. Third, the full texts of these articles were read, and 3 articles were eliminated; another article was discarded because the full text was not available. Finally, 8 articles were ultimately included in the meta-analysis.

## 3. Results

### 3.1. Characteristics of the Studies

The total sample size was 952 patients. The age range of the participants in the selected studies was between 40 and 70 years. All the RCTs were carried out in different countries: Spain, Iran, Beijing, Sweden, the USA, Finland, the United Kingdom, and Poland. HRQOL was measured with the 15-item Cervantes scale and the SF-36, WHQ, and SF-12 questionnaires. Physical activity interventions had an average duration of 30–60 min per session, with a frequency of 3–4 sessions per week over a period of 12–15 weeks (which is the minimum recommended duration for assessing the effects of physical exercise on participants’ quality of life). The intervention sessions generally included three phases: warm-up, either AE (walking or Nordic walking) or RT (weight machines and elastic bands), and muscle stretching. The analysed studies investigated the effects of different physical training modalities on VMSs, insomnia, and physical and mental dimensions of women in climacteric. The main characteristics of the studies included in the review and meta-analysis are summarised in Table 1 [16,19,20,28,29,30,31,32].

### 3.2. Quality Assessment

Overall, four (50%) of the studies had a low risk of bias, two (25%) had an unclear risk of bias, and the remainder (25%) had a high risk of bias (Figure 2). Most studies had an unclear risk of bias concentrated in performance bias (blinding participants and staff). However, most RCTs reported good adherence to the intervention.

### 3.3. Effects of AE on Climacteric Symptoms and Quality of Life

The effects of AE on climacteric symptoms and QoL were evaluated in five studies, all of which revealed significant changes in the time domain; that is, the duration of the intervention was a crucial factor in assessing long-term health changes in women [20,28,29,30,31,32]. The five trials included 260 randomised women who were perimenopausal, menopausal, or postmenopausal without serious somatic or psychological disorders that could interfere with exercise. In addition, the women did not necessarily exercise regularly and presented symptoms of menopause, such as anxiety, insomnia, depression, or VMSs, such as five episodes of hot flashes/night sweats per day.

In two of the trials, VMSs were one of the inclusion criteria [16,30]. RCTs with a placebo group indicated that behavioural interventions, such as daily AE, including walking and breathing exercises, may be effective in alleviating climacteric symptoms and increasing QoL. Zhang et al., (2014) reported that, after 12 weeks of AE, the women in the intervention group exhibited decreases in the total Kupperman index, a test that classifies climacteric symptoms [34], and other parameters, such as body weight, waist contour, body mass index, and triglyceride and total cholesterol levels, compared with the baseline values [28]. Abedi et al. (2015) reported that walking with a pedometer for 12 weeks, increasing the number of steps each week, had a beneficial effect on mood, sleep disorders, and irritability among the postmenopausal women in the study [29]. Similar to other behaviour change programs, a web-based program to improve the experience of menopausal symptoms for middle-aged women (WPAPP-M) that incorporated AE, support and information to encourage physical exercise and manage menopausal symptoms was used in one study. This study demonstrated that more than 3 months of exercise was needed to significantly improve menopausal symptoms, although the VMSs gradually decreased over time [30]. In contrast, a UK trial investigating the efficacy of AE as a treatment for menopausal VMSs at 6 months concluded that exercise was not an effective treatment for hot flashes and night sweats [20]. Due to these differing results, the effects of physical exercise on VMSs are controversial. The results of most of the trials suggest that controlled and regular AE positively affects the vitality and psychological health of women; therefore, their inclusion in a healthy lifestyle leads to an improvement in QoL [31].

### 3.4. Effects of RT on Climacteric Symptoms and Quality of Life

A total of three RCTs investigated the effects of RT on the symptoms and QoL of 125 perimenopausal, menopausal, and postmenopausal women [16,19,31]. The FLAMENCO project investigated the influence of a physical training program with RT and other exercises for 16 weeks. At the end of the intervention, the abbreviated 15-item Cervantes scale was used to evaluate HRQOL during menopause. The women in the exercise group had better scores related to self-perception of health, decreased vaginal moisture, relationships, and mental well-being than the women in the control group did. In addition, the loss of bone mineral density was slowed and bone density was increased, especially in the pelvis, compared with the control group [16]. In the trial by Berin et al. (2021), in postmenopausal women in the intervention group, the WHQ scores showed improvements in the domains of VMSs, sleep problems, and menopausal symptoms after a 15-week RT intervention period [19]. In another trial of menopausal women, after 12 weeks of training including warm-up exercises, strengthening with elastic bands and cool-down exercises (3 times/week for 60 min), a positive difference was observed in all domains of the SF-36 questionnaire in the intervention group. This study revealed a significant change in the vitality and mental health of women [31]. Thus, RT is especially beneficial during climacteric, as the exercise helps to strengthen muscles and increase bone density, preventing osteoporosis and improving QoL in women [35].

### 3.5. Meta-Analysis Results

Of the studies in the systematic review, five trials included data for the AE meta-analysis (n = 374 in the AE group and n = 338 in the control group), and three included data for the RT (n = 132 in the RT group and n = 125 in the control group). The relative risk (RR) for the occurrence of health events was RR = 1.13 (95% confidence interval [CI]: 1.02–1.26) in favour of the AE group, but this difference was not statistically significant (Figure 3). The relative risk was similar (RR = 1.06 [95% CI: 0.89–1.26]) for the RT group. As shown in both forest diagrams, the heterogeneity of the studies was low according to the I^2^ values, indicating that the differences between the studies are minimal and are likely attributable to chance. In addition, both graphs showed that the consistency between the studies was high (Figure 4).

## 4. Discussion

Three groups of subjects were identified in the included studies: women in premenopause [20,28], women in menopause [16,30,31,32], and women in postmenopause [19,29]. Menopause received the most attention in the studies analysed in this systematic review likely because of the significant physiological changes that occur at this time in a woman’s life; however, the importance of studying the early and later stages of menopause is evident, because these stages can also influence long-term health and QoL [12]. Women who received any type of hormonal or natural treatment were excluded from the studies, since these treatments can modify physiological and hormonal parameters of interest, such as changes in muscle mass, body composition, or physical performance, which could influence the validity of the results [13,14].

According to Zhang et al. (2014), AE, such as walking with long strides, is an effective method for improving most symptoms of perimenopausal syndrome. After 12 weeks of training three times per week or more, women in the intervention group experienced improvements in insomnia, irritability, joint or muscle pain, and fatigue [28]. In the study by Daley et al. (2015), perimenopausal women performed 30 min of moderate-intensity AE at least three days per week (including brisk walking, jogging, aerobics, swimming, cycling, and tennis). After six months of practising AE, significant improvements in sleep were observed, although hot flushes and night sweats did not improve [20]. During menopause, Mansikkamäki et al. (2015) reported physical and mental improvements after 12 weeks of AE training [30]. Similar results were obtained by Eun-Ok Im et al. (2017), who, after three months of AE, reported improvements in psychological symptoms in menopausal women [32]. In the same vein, Abedi et al. (2015) concluded that AE has positive effects on depression, insomnia, and anxiety in postmenopausal women [29]. These studies conclude that performing AE regularly during perimenopause, menopause, and postmenopause predominantly benefits psychological aspects (depression, insomnia, anxiety, and improved sleep). AE provides fewer physical improvements and does not influence the improvement of VMSs.

Regarding research on the effects of RT, Berin et al. (2021) studied its benefits in postmenopausal women. After 15 weeks of structured RT in two to three weekly sessions, improvements were observed in sleep quality and VMSs [19]. Similarly, Baena et al. (2022) reported that after implementing an RT programme consisting of 60-minute sessions, three days per week for 16 weeks, benefits were observed not only in VMSs but also in relationship quality and mood state [16]. Likewise, Dąbrowska et al. (2016) evidenced that regular RT over 12 weeks (three times per week, 60 min per session) led to positive changes in vitality and mental health in menopausal women [31]. It is important to highlight that RT has a significant positive impact on improving psychological symptoms and VMSs, in addition to increasing vitality during the climacteric phase.

The results of the meta-analysis indicate that physical exercise performed on a regular basis, including AE and FE RT, can reduce the severity of climacteric symptoms such as emotional lability, insomnia, and fatigue. This reduction is associated with benefits in the general perception of health, which leads to an increase in HRQOL. The benefits of physical exercise may depend on the characteristics, intensity, and duration of the exercise, as well as on individual factors, such as the general state of health and hormonal levels of the women [17].

All the trials used interventions with a minimum duration of 12 weeks. The Spanish Association for the Study of Menopause (AEEM) recommends a training plan of 8–12 weeks of physical exercise as the minimum time to achieve the effects on the health of women in climacteric [36]. The exercise programs recommended for postmenopausal women according to the American Health Association (AHA) and the American College of Sports Medicine (ACSM) are divided into three areas: AE (walking, swimming, tennis, dancing and cycling) for 30 min, five times/week at a moderate intensity or 20 min three times/week at a vigorous intensity; muscle strengthening performed two times/week on non-consecutive days focusing on large muscle groups (legs, arms, shoulders, abdomen and hips); and elasticity and balance exercises to improve stability and decrease the risk of falls [37].

Although regular exercise is essential in climacteric, many women lack access to adequate guidance and comprehensive support to take full advantage of its benefits [38], which indicates a significant gap in knowledge.

Interpretation of our findings requires consideration of several limitations. First, the limited number of eligible RCTs available on the subject affects the generalisability and statistical power of our meta-analysis results. Furthermore, the small sample sizes in most trials [18,39] resulted in less precise estimates of effects. This limitation must be acknowledged, as it prevents the establishment of definitive guidelines based on the current data. It would be interesting to develop specific clinical exercise guidelines for women in the climacteric period, adapted to different ages, physical conditions, and predominant symptoms. As well as the incorporation in health services of exercise prescription as part of the follow-up of these women. Future research should involve larger samples of women at various stages of menopause to determine the most effective exercise interventions in climacteric. Specifically, studies should focus on comparing the efficacy of RT and AE, investigating the optimal duration and intensity of each type of exercise, and exploring the role of patient-reported outcomes. The lack of conclusive evidence on the effects of exercise on VMSs highlights the need for further research to clarify the potential benefits of exercise. The findings of this study confirm that regular physical exercise has a positive impact on the QoL of women during the climacteric period. However, our results also reflect variability in the response to physical activity, particularly concerning the reduction in VMSs. While Daley et al. [20] suggest that AE may not be effective in treating VMSs and night sweats, other research has reported that these symptoms decrease following the application of RT and resistance [16,19].

Additionally, adherence to exercise programmes remains a key challenge, highlighting the need to develop personalised and motivational strategies to promote sustained implementation. In this regard, hybrid interventions that combine supervised exercise with digital or telemedicine tools could facilitate access and long-term adherence [16].

Furthermore, the present study highlights that both aerobic and resistance training provide significant benefits, albeit with distinct effects. While aerobic exercise promotes cardiovascular health and body weight control, resistance training contributes to the prevention of sarcopenia and the improvement of bone density, which are fundamental aspects at this stage of life [14].

## 5. Conclusions

The findings of this systematic review and meta-analysis reinforce the notion that physical exercise is an effective tool for improving the quality of life in women during the climacteric period, highlighting its benefits for psychological well-being, metabolic health, and bone health. However, the heterogeneity of the observed effects on the reduction in VMSs suggests the need for further studies to determine which exercise modalities, intensities, and durations may be most effective in this context. Future research should focus on evaluating long-term exercise programmes with larger sample sizes and standardised methodologies. Additionally, it would be advisable to explore strategies to enhance adherence to physical activity in this population, incorporating personalised approaches and technological tools to facilitate monitoring and motivation. 

Finally, although exercise is presented as a safe and accessible alternative, it is essential that recommendations on physical activity for women during the climacteric period are integrated within a comprehensive care approach, encompassing both nonpharmacological interventions and conventional therapeutic options when necessary.

## Figures and Tables

**Figure 1 healthcare-13-00644-f001:**
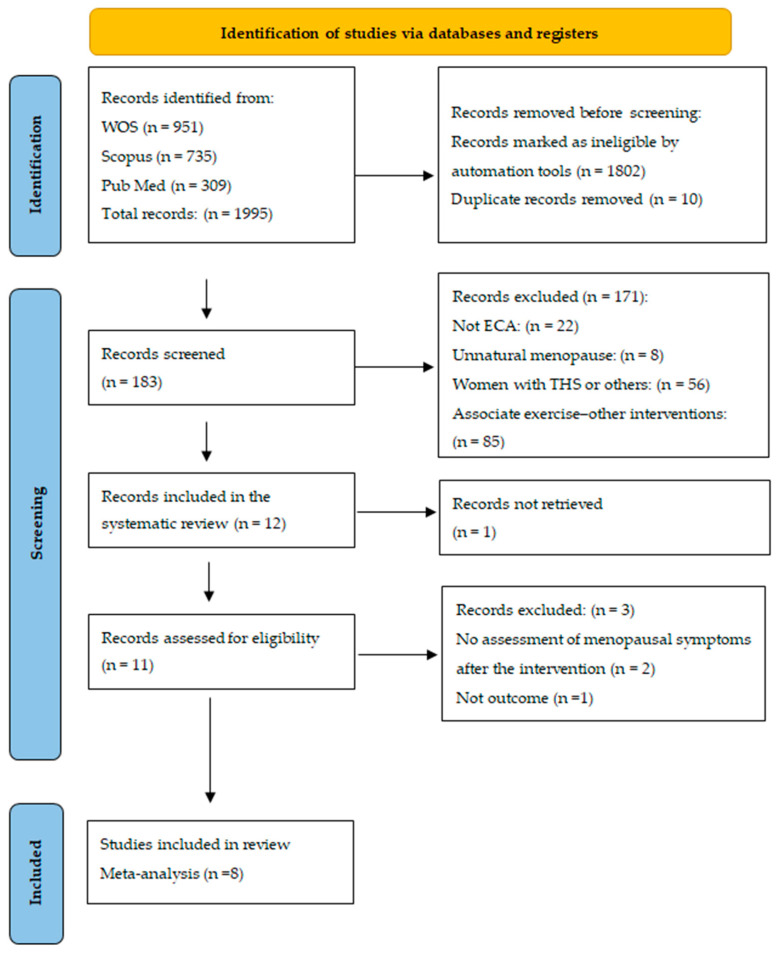
Flowchart of the search strategy according to PRISMA.

**Figure 2 healthcare-13-00644-f002:**
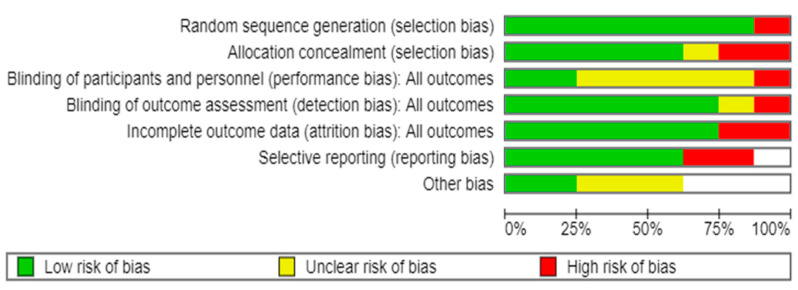
Risk of bias assessment of individual studies included in the systematic review and meta-analysis [16,19,20,28,29,30,31,32].

**Figure 3 healthcare-13-00644-f003:**
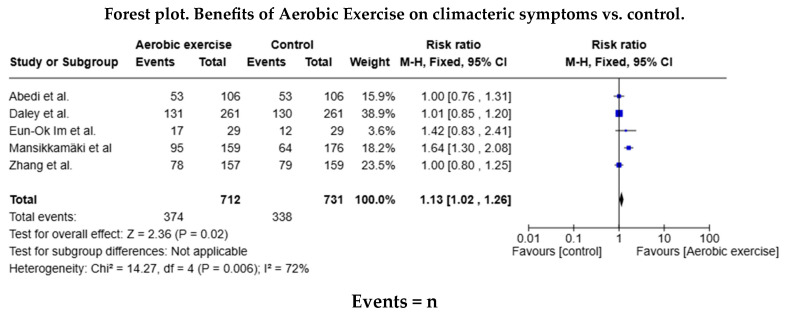
Forest diagram for aerobic exercise vs. control [20,28,29,30,31,32].

**Figure 4 healthcare-13-00644-f004:**
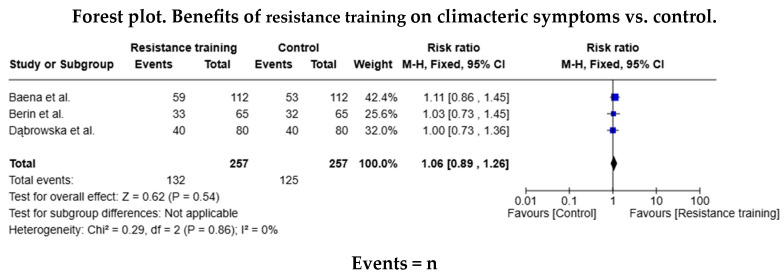
Forest diagram for resistance training vs. control [16,19,31].

**Table 1 healthcare-13-00644-t001:** Characteristics of the studies included in the systematic review and meta-analysis.

Study	Participants and Climacteric Stage	Groups	Interventions/Times	HRQOL Measurement	Outcomes
Daley et al., 2015 [20] United Kingdom	n = 261 Age = 48–57 years Perimenopause Postmenopause	Experimental group: (n = 131) Control Group: (n = 130)	Experimental Group: AE (30 min 3 days/week) Time: 6 months Control Group: social support for exercise	WHQ: measures women’s health SF-12: QoL	Aerobic exercise improves women’s quality of life but does not reduce hot flushes and night sweats
Zhang et al., 2014 [28] Beijing	n = 157 Age = 40–55 years Perimenopause	Experimental group: (n = 78) Control group: (n = 79)	Experimental group: AE (walking 4000 steps) Frequency: 3 times/week Duration: 30 min/session Time: 12 weeks Control group: usual care	Kupperman Index: assesses the severity of menopause-related symptoms	AE substantially reduces menopausal symptoms
Abedi et al., 2015 [29] Iran	n = 106 Age = 40–60 years Postmenopause	Experimental group: (n = 53) Control group: (n = 53)	Experimental group: AE (walking with a pedometer, increasing by 500 steps/week) Time: 12 weeks Control group: usual care	GHQ28 and BECK: measures anxiety, depression and insomnia	AE decreases effects on depression, insomnia, and anxiety
Mansikkamäki et al., 2015 [30] Finland	n = 159 Age = 40–63 years Menopause	Experimental group: (n = 95) Control group: (n = 64)	Experimental group: AE (walking or Nordic 2 days a week) Frequency: 4 times/week Duration: 50 min Time: 12 weeks Control group: habitual physical activity	SF-36: measure QoL	Improves physical and mental dimensions of quality of life
Dąbrowska et al., 2016 [31] Poland	n = 80 Age = 40–65 years Menopause	Experimental group: (n = 40) Control group: (n = 40)	Experimental group: RT Frequency: 3 times/week Duration: 60 min Time: 12 weeks Control group: habitual physical activity	SF-36: measures QoL	Positive change in vitality and mental health
Eun-Ok Im et al., 2017 [32] United States	n = 29 Age = 40–60 years Menopause	Experimental group: (n = 17) Control group: (n = 12)	Experimental group: AE (training and information program to increase physical activity) Time: 3 months Control group: no access to the program	MSI: measures psychic, psychosomatic and physical symptoms Kaiser Physical Activity Survey: measures physical activity	Improves menopause symptoms + mood and physical endurance
Berin et al., 2021 [19] Sweden	n = 65 Age = 45–70 years Postmenopause	Experimental group: (n = 33) Control group: (n = 32)	Experimental group: RT (muscle strengthening and stretching) Time: 15 weeks Control group: usual care	WHQ: measures women’s health SF-36: measures QoL	Improves postmenopausal symptoms (hot flushes and night sweats)
Baena et al., 2022 [16] Spain	n = 112 Age = 45–60 years Menopause	Experimental group: (n = 59) Control group: (n = 53)	Experimental group: RT (balance training, muscle strengthening) Frequency: 3 times/week Duration: 60 min Time: 12 weeks Control group: 4 talks on the health benefits of physical exercise and the Mediterranean dietary pattern	Cervantes Scale: measures HRQoL in menopause	Improves menopause symptoms, relationship quality, mood, and VMSs

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
