# Peer review of "Effects of Physical Exercise on Symptoms and Quality of Life in Women in Climacteric: A Systematic Review and Meta-Analysis"

_healthcare, 2025, doi:10.3390/healthcare13060644_

Round 1
Reviewer 1 Report (Previous Reviewer 1)
Comments and Suggestions for Authors
The authors have greatly improved the text. There are only a few abbreviations where the word and its abbreviation are mentioned more than once, which should be correct on the first occasion only. There should be no problem with publishing it after this correction.
Author Response
We appreciate your comments and your positive assessment of the manuscript. We have corrected unnecessary repetitions of abbreviations, ensuring they are defined only upon their first occurrence. Thank you for your review and your valuable suggestions.

Reviewer 2 Report (Previous Reviewer 2)
Comments and Suggestions for Authors
Many thanks to the editor for the submission of this study.
The article is an interesting literature review that examines the effect of different types of physical activity on a sample that deserves increasing attention.
Below, I provide some comments that, in my opinion, would help improve the article:
Lines 41-42: I would suggest rewording this sentence for better clarity.
Line 67: It seems that, possibly due to language issues, you have abbreviated “resistance training” as FE. While this might be acceptable, you later start using RT (from line 283 onwards). I suggest maintaining this latter abbreviation, as it is the most commonly used in literature.
Line 136: search equation - - > search string
Lines 152-158: Which table are you referring to? It might seem like a description of Table 1, but the sections do not match those described.
Line 289: “Vaginal problems”? What exactly do you mean?
Lines 375-378: Which studies state that there are no VMS effects, and which ones claim the opposite? Among the studies that reported positive outcomes, what type of physical activity was proposed? It would be interesting to understand which type of physical activity yielded the most beneficial effects.
Table 1: In my opinion, this is the main issue of the article. The outcome column should include much more quantitative information reported by the authors of the various studies (e.g., differences between pre- and post-training). While this may be acceptable for a narrative review, a systematic review and meta-analysis require a greater amount of data.
Figures 3, 4: The data presentation does not seem very intuitive. What do you mean by “events”? Perhaps you could clarify this in the paragraph or in the legend to improve the understanding of the tables.
I hope this comments could help the autors
Author Response
We greatly appreciate your comments and suggestions, which have been extremely helpful in improving the clarity and quality of our manuscript. Below, we outline the changes made in response to your observations:
- Lines 41-42: We have sought to provide a clearer explanation of the concepts of menopause, early menopause, and late menopause.
- We have corrected the error and standardised all mentions of 'resistance training' under the abbreviation RT, following your recommendation and the most common usage in the literature.
- Lines 152-158: The table was created with the aim of organising the information extracted from the reviewed articles. Subsequently, this information was restructured in Table 1, to which the corresponding results were added for a clearer and more complete presentation.
- Line 289: "vaginal problems" has been changed to "decreased vaginal moisture".
- Lines 375-378: This part of the discussion has been improved and clarified.
- Table 1: has been completed.
- The legend has been updated. The figures could not be modified as they were generated by Review Manager 5.4 software.
Once again, we sincerely appreciate your thorough review and your valuable recommendations, which have significantly contributed to improving our work. We remain open to any further suggestions you may have.

Reviewer 3 Report (New Reviewer)
Comments and Suggestions for Authors
Reviewer Comments:
- Abstract
- Relate the research problem to the objective of the study. Keywords
- Introduction
- What other non-pharmacological treatments are scientifically proven to be of benefit in this research topic?
- Can there be differences between the two phases of the climacteric in terms of the benefits of conservative treatments? The type of exercise may be different?
- To make a greater contextualisation of the type of physical exercise, dosage, intensity... and the benefits in the climacteric period, according to the available evidence.
- Methods
- Table 1: In some studies the rating scale is not indicated.
- Results
- The differences in the benefits of exercise at different stages in terms of type of exercise, dosage, intensity, frequency, etc. should be clarified.
- Dicussion and Conclusion
- Discuss the results of the benefits of exercise at different stages in terms of type of exercise, dosage, intensity, frequency... as well as their evaluation using the scales analysed.
- specify the types of physical exercise with the greatest benefits. Not only in a general way ‘physical exercise’.
Author Response
We sincerely appreciate your observations and the time you have dedicated to reviewing our manuscript. Your comments have been instrumental in enhancing the accuracy and clarity of our work. Below, we detail the changes made in response to your suggestions:
- Abstract: A linking sentence has been added between the research problem and the objective.
- Introduction: We acknowledge that there are other non-pharmacological alternatives recommended during this stage of life; however, these do not form part of the specific objective of our research. A sentence listing these alternatives has been added.
- Regarding the benefits of physical exercise, its type, dosage, intensity, and frequency, these have been detailed in the Results section, based on the available evidence. However, we will consider your suggestion for future, broader systematic reviews, which will also take into account other factors such as diet and rest.
- Table 1: All assessment scales have been added.
- Results: Details have been incorporated into Table 1.
- The benefits of exercise at different life stages have been discussed in terms of exercise type, dosage, intensity, and frequency, as well as its evaluation using the relevant assessment scales.
- The types of physical exercise with the greatest benefits have been specified.
We sincerely thank you again for your detailed review and your valuable suggestions, which have significantly enriched our study. We remain at your disposal for any further recommendations you may consider necessary.

Round 2
Reviewer 2 Report (Previous Reviewer 2)
Comments and Suggestions for Authors
I thank the editor for assigning me the manuscript for review, and I thank the authors for the revisions made following my comments.
The manuscript has been significantly improved after the first revision.
All the suggested changes have been implemented. With this new version, I have no further comments. I believe it is ready for publication.
Author Response
Dear Reviewer,
I am pleased to hear that you consider the quality of the manuscript to have improved following the incorporation of your suggestions.
I greatly appreciate the time and effort you have dedicated to reviewing my work and providing constructive feedback that has contributed to its enhancement. Your insights have been instrumental in strengthening the clarity and robustness of my research.
Once again, thank you for your help and for your commitment to improving the article.
Reviewer 3 Report (New Reviewer)
Comments and Suggestions for Authors
Most of the reviewer's suggestions have been addressed.
Author Response
Dear Reviewer,
I am pleased to hear that you consider the quality of the manuscript to have improved following the incorporation of your suggestions.
I greatly appreciate the time and effort you have dedicated to reviewing my work and providing constructive feedback that has contributed to its enhancement. Your insights have been instrumental in strengthening the clarity and robustness of my research.
Once again, thank you for your help and for your commitment to improving the article.
This manuscript is a resubmission of an earlier submission. The following is a list of the peer review reports and author responses from that submission.
Round 1
Reviewer 1 Report
Comments and Suggestions for Authors
The article is very interesting due to the importance of the parameters analyzed (physical exercise on symptoms and quality of life) and the stage of life of the woman analyzed. Nevertheless, there are several points that were not touched upon during the text that need to be mentioned.
In the introduction, there is a brief mention of physical and mental situations of menopause, which are basic in this stage of women's life, and especially for the study. On the other hand, pharmacological interventions to compensate for the hormonal decrease, which are the most common treatments, are also not mentioned at any time.
Likewise, the authors make a very adequate breakdown of the PICO question, however, during data collection, and the risk of bias is not mentioned who developed the risk analysis, as well as the PRISMA is an image and not editable, the number of articles found in the identification section (1995) does not correspond with the number of those eliminated due to duplicates (n = 10), those automatically eliminated (193) with 183 those searched for to perform the screening (1995- (10 + 193) = 1792). On the other hand, later it is not mentioned why 171 of the articles were excluded before the reports sought for retrieval. Was it for not mentioning some of the topics sought? Was it for including non-specific parameters? Was it for not being randomized studies? Or was it because some studies mentioned drug treatments and so they were excluded?
Due to the above mentioned, the results, discussion and conclusions should be verified because there are doubts that are important for the reader. Regarding minor situations, the bibliography is not in accordance with the request of the publisher and the patent section that was left by mistake should be eliminated.
Author Response
RESPONSE TO REVIEWER 1
We sincerely appreciate the time and dedication invested in reviewing our manuscript. In response to your observations, we have made the following modifications:
- To address this point regarding the physical and mental conditions during menopause, we have added a more detailed description of the symptoms and their impact on women's overall health. Additionally, we acknowledge the omission of pharmacological interventions, which are a widely used treatment option to counteract the effects of hormonal decline. Therefore, we have incorporated a reference to hormone therapies and other pharmacological treatments used in menopause, highlighting their role in alleviating symptoms and comparing them with physical exercise as a non-pharmacological strategy.
- Regarding your observation on the need to clarify who conducted the risk of bias analysis during data collection, and in order to enhance the transparency of the study, we have added this information in the corresponding section.
- Following your comment on the error in developing the PRISMA algorithm, we have removed the image to make it editable and corrected the relevant calculations. Furthermore, we have specified the reasons for excluding the eliminated articles.
- In response to your observations regarding the need to review the results, discussion, and conclusions to ensure greater clarity and precision in the interpretation of the findings, we have conducted a thorough verification of these sections. We have ensured that the results are presented consistently with the methodology used and that the discussion appropriately contextualises the findings within the existing literature. Likewise, we have revised the conclusions to reinforce the relationship between the findings and their clinical applicability.
- Regarding formal aspects, we have adjusted the references to comply with the journal’s requirements, ensuring the correct application of the requested format. Additionally, we have removed the patents section, as it was included by mistake.
Once again, we appreciate your comments, as they have contributed to improving the rigour and clarity of the manuscript.

Reviewer 2 Report
Comments and Suggestions for Authors
Many thanks to the editor for sending me this article to review.
The article appears clear and well written regarding the rationale and methods used. The critical points have already been addressed by the authors.
The sample size is too small to establish guidelines. Additionally, the review was not registered in a systematic review database and was conducted independently. Although these two aspects are mentioned by the authors, they should be included in a specific section dedicated to the study's limitations.
line 185: with "...0–3 sessions performed per week.". Does this mean that some weeks had zero sessions?
Table 1: In addition to the sample size and the volume of physical activity, I would highlight the type of activity performed by the intervention group (aerobic, resistance, or mixed) and add a section summarizing the study outcomes.
Author Response
RESPONSE TO REVIEWER 2
We sincerely appreciate the time and dedication invested in reviewing our manuscript. In response to your observations, we have made the following modifications:
- We appreciate your comment regarding the sample size and agree that this is a limitation to be considered when interpreting the results. In the limitations section, we had already noted that the limited number of randomised controlled trials (RCTs) included affects the generalisability and statistical power of the meta-analysis. We also mentioned that the small sample sizes in most trials may have resulted in less precise estimates of the effects of exercise. However, to emphasise this aspect and address your observation more clearly, we have made a minor adjustment in this section, highlighting that this limitation prevents the formulation of definitive guidelines based on the current data.
- Regarding your observation about the lack of registration in a systematic review database such as PROSPERO, we would like to clarify that we had considered this registration. However, we decided to wait for the journal's decision to ensure that the study passed the initial quality checks and met the necessary criteria for inclusion in such a database. Despite not being registered beforehand, we want to emphasise that our review rigorously followed the PRISMA 2020 guidelines, ensuring transparency and reproducibility of the process. Moreover, all methodological procedures were detailed in the manuscript to guarantee the study's traceability.
- Following your observation regarding the number of sessions in the physical interventions, we have modified the text to clarify that the number of sessions ranges between three and four per week. Similarly, we have made adjustments to the duration of these sessions to avoid possible misunderstandings.
- In Table 1, we have added the type of activity for each intervention group and expanded the table with an additional column summarising the results of each study.
Once again, we appreciate your comments, as they have contributed to enhancing the rigour and clarity of the manuscript.

Reviewer 3 Report
Comments and Suggestions for Authors
The manuscript entitled “Effects of physical exercise on symptoms and quality of life in women in climacteric: A systematic review and meta-analysis” presents interesting issues however some questions arise.
- Please explain (in the review, not in the article) why. Registering in a database is a very important aspect of conducting a systematic review and meta-analysis, as it helps standardize certain aspects and verify whether similar systematic reviews and meta-analyses have already been conducted.
- The flow chart should be in the methodology, not in the results. It is necessary to clearly describe the procedures for selecting and excluding articles. The flowchart states that 193 were automatically rejected—what tool was used for this? Additionally, it is important to specify how many people participated in each stage. What happened when two reviewers had differing opinions?
- The first sentence in the results section and the flowchart do not match regarding the number of rejected articles (183 vs 193). This must be checked and clarified.
- The flowchart shows how many publications were included in the meta-analysis, but what about the systematic review?
- On the flowchart, "Record" appears once and "Reports" at another point—these are two different things. Please check the original document. Please correct this.
- In the subsection "Effects of AE on climacteric symptoms and quality of life," the font size changes. Please correct this.Dół formularza
- There is no strict universally accepted minimum of number of publications for a meta-analysis but many guidelines suggest including at least 3 to 5 studies. Three publications are the threshold value, and it can be problematic. More publications help provide a more reliable and valid meta-analytic conclusion
- “This systematic review suggests that RT can help prevent sarcopenia and improve bone health, potentially reducing the risk of osteoporosis, which is often associated with menopause, by stimulating bone mineral density and increasing bone strength” - The first conclusion does not directly follow from the data presented in the article. A table summarizing these results from articles included in systematic review should be added (for example, there is no mention of sarcopenia, except in the introduction). The results should only be related to the analysis of the articles (systematic review) and the meta-analysis – there should be no recommendations (recommendations can be included in the discussion). Please focus the results solely on what has been determined (e.g., in the meta-analysis).
- In the systematic review, there is only one table that provides information about the intervention (type of intervention and its duration) and the tool used to assess quality of life. However, the conclusions are very broad and refer to sarcopenia, bone health, potentially reducing the risk of osteoporosis, cardiovascular health and weight control, and a reduction in depression – these were not the subject of this analysis and are an overstatement (they were not included in the keywords). Either more results need to be added in the tables (and subjected to meta-analysis – if possible, as they are likely different outcomes), or this should be removed (since there is no such data from this systematic review and meta-analysis, and it was not considered in the keywords for the search, so some related publications may not have been included in this article).
Author Response
RESPONSE TO REVIEWER 3
We sincerely appreciate the time and dedication invested in reviewing our manuscript. In response to your observations, we have made the following modifications:
- Regarding your comment on the importance of registering systematic reviews and meta-analyses in databases, we acknowledge that registration on platforms such as PROSPERO is a recommended practice, as it helps ensure process transparency, minimises the risk of publication bias, and prevents unnecessary duplication. In this case, we decided not to register the review beforehand as we were awaiting editorial evaluation to confirm that we met the necessary methodological standards before considering its inclusion in a database. Nonetheless, we would like to emphasise that the study strictly followed the PRISMA 2020 guidelines, ensuring a rigorous and transparent methodology. Furthermore, we have clearly detailed the search process, study selection, and data analysis in the methods section, allowing for the replicability of our work. We understand the significance of prior registration and will consider implementing this in future systematic reviews to further enhance the quality and standardisation of our studies.
- Following your observation regarding the placement of the PRISMA algorithm in the review, we have moved it to the methodology section as suggested. Additionally, we have specified the reasons for excluding the eliminated articles and corrected the transcription error in the data. Regarding your question on what happened when two reviewers had differing opinions, we clarify in line 144 that any disagreements were resolved with the involvement of a third and fourth author. We have also specified the articles included in the systematic review and corrected the misuse of the terms “report” and “record.”
- Concerning your comment on the change of font in the subsection Effects of Exercise Interventions on Climacteric Symptoms and Quality of Life, we have corrected the font and reviewed the entire manuscript to ensure no other formatting inconsistencies remain.
- With respect to your remark on the minimum number of articles recommended by guidelines for conducting a meta-analysis, it should be noted that, after rigorously applying the search equations and strictly considering the inclusion and exclusion criteria, a total of eight articles were included. Although this number is limited, we believe it is sufficient to derive reliable and valid conclusions.
- Following your comment on presenting the first conclusion with data directly extracted from the study, since sarcopenia is only mentioned once in the introduction as another symptom of menopause, we realised that, in translation, where we abbreviated resistance training as (EF), it was mistakenly written as (RT), leading to confusion with radiotherapy. We have therefore corrected this abbreviation throughout the manuscript.
- In the results section, we have focused on analysing the articles included in the meta-analysis and moved the recommendations to the discussion section.
- In the conclusions section, we have removed references to sarcopenia, bone health, the potential reduction of osteoporosis risk, cardiovascular health, and weight management, as these aspects were not part of the analysis of the selected studies. Instead, we have formulated a conclusion that is more aligned with our review.
Once again, we appreciate your comments, as they have contributed to improving the rigour and clarity of the manuscript.
